# Optimization of COVID-19 vaccination and the role of individuals with a high number of contacts: A model based approach

Tarcísio M. Rocha Filho[1,2]*, José F. F. Mendes[3], Thiago B. Murari[4], Aloísio S. Nascimento Filho[4], Antônio J. A. Cordeiro[4,5], Walter M. Ramalho[6], Fúlvio A. Scorza[7], Antônio-Carlos G. Almeida[8], Marcelo A. Moret[4,9]

1 International Center for Condensed Matter Physics, Universidade de Brasília, Brasília, DF, Brazil, 2 Instituto de Física, Universidade de Brasília, Brasília, DF, Brazil, 3 Departamento de Física & I3N, Universidade de Aveiro, Aveiro, Portugal, 4 Centro Universitário SENAI CIMATEC, Salvador, BA, Brazil, 5 Instituto Federal de Educação e Tecnologia da Bahia, Feira de Santana, BA, Brazil, 6 FCE and Núcleo de Medicina Tropical, Universidade de Brasília, Brasília, DF, Brazil, 7 Disciplina de Neurociência, Escola Paulista de Medicina/ Universidade Federal de São Paulo (EPM/UNIFESP), São Paulo, SP, Brazil, 8 Universidade Federal de São João del-Rei (UFSJ), São João del-Rei, MG, Brazil, 9 Universidade do Estado da Bahia, Salvador, BA, Brazil

* marciano@fis.unb.br

**Data Availability Statement:** All relevant data are within the manuscript and its Supporting information files.

## Abstract

We report strong evidence of the importance of contact hubs (or superspreaders) in mitigating the current COVID-19 pandemic. Contact hubs have a much larger number of contacts than the average in the population, and play a key role on the effectiveness of vaccination strategies. By using an age-structures compartmental SEIAHRV (Susceptible, Exposed, Infected symptomatic, Asymptomatic, Hospitalized, Recovered, Vaccinated) model, calibrated from available demographic and COVID-19 incidence, and considering separately those individuals with a much greater number of contacts than the average in the population, we show that carefully choosing who will compose the first group to be vaccinated can impact positively the total death toll and the demand for health services. This is even more relevant in countries with a lack of basic resources for proper vaccination and a significant reduction in social isolation. In order to demonstrate our approach we show the effect of hypothetical vaccination scenarios in two countries of very different scales and mitigation policies, Brazil and Portugal.

## Introduction

The first cases of human transmission of SARS-CoV-2 were reported in Wuhan Province in China in December 2019 [1]. By January 2020, the spread became an epidemic and was declared a pandemic on March 11 by the World Health Organization [2]. Since then, the virus has spread over all countries in the world, with more than 208 million total cases and 4.3 million deaths [3]. The estimates for the basic reproduction number $R_0$ and herd immunity are estimated in the range of 2.8–3.3 [4, 5] and 0.64–0.7 [6, 7], respectively. With an infection fatality rate of 0.657% [8] (that varies according to the age distribution in the

**Funding:** This work received financial support from the National Council of Technological and Scientific 242 Development - CNPq (grant numbers 302449/ 2019-1 FAS, 309617/2020-0 ACGA, 243 305291/ 2018-1 MAM), Bahia State Research Support Foundation (BOL0723/2017 AJAC) 244 (Brazil) and i3N (grant numbers UIDB/50025/2020 & UIDP/ 50025/2020) - Fundação para a 245 Ciência e Tecnologia/MEC (Portugal). The funders had no role in study design, data collection 246 and analysis, decision to publish, or preparation of the manuscript.

**Competing interests:** The authors have declared that no competing interests exist.

population), the natural free evolution would imply a death toll too large to be even considered as a possibility, and would result in overcrowded medical facilities, and an even larger economic impact than the one endured up to the present time [8–11]. Besides that, the duration of disease-generated immunity is not yet well known, with the complicating factor that the free circulation of the SARS-CoV-2 has lead to new potentially dangerous mutations [12, 13]. As a consequence, the different vaccines developed up to the current date [14] are important tools for effectively mitigating the current COVID-19 pandemic. An efficient and properly designed immunization strategy would certainly result in the best payoffs, for the whole health system, for the population well-being, and resulting to something close to a fully working economy.

The World Health Organization COVAX initiative, a global vaccine alliance, aims to allocate two billion vaccine doses during 2021 across different participant countries [15], roughly a quarter of the world population. A total of 4.7 billion vaccine doses were administered in the world [3] to the current date for a total population of 7.8 billion, with a limited number of vaccine doses available, especially in poorer countries. As a consequence priorities were established for the vaccination order, mainly by age and healthcare personnel, workers in essential and critical industries, individuals at higher risk for severe COVID-19, and individuals of 65 years of age and older [16]. In the European Union elderly people, healthcare workers and individuals with certain comorbidities were the first in line [17]. In Brazil, third in number of cases and second in deaths among all countries, the vaccination just started with healthcare workers, 75 years of age and older individuals, long-term care facilities patients with 60 years of age and older, indigenous peoples living in reservations, and traditional communities in river banks, individuals with comorbidities, and now reaching younger adults [18].

A priority ranking becomes unavoidable as the selected priority groups may constitute a considerable fraction of the population. This may change with time as one expects the general guidelines to be modified as more evidence is gathered on the COVID-19 epidemiology and on the vaccine safety and efficacy for each target group. A survey carried out in Belgium asked 2060 participants aged 18 to 80 who should be vaccinated first, second and so on and reached no consensus [19], showing that the perception of such priorities in the whole population is not clear yet. A methodology to carefully analyze the current situation, considering the social contacts structure in a given population and its age distribution, is necessary in order to design the most efficient approach for the vaccination against COVID-19, in order to minimize deaths, hospitalizations and other negative impacts of the current pandemic.

The stage of the pandemic in each country must also be considered, as some vaccine variants may be more effective in reducing the likelihood of severe COVID-19 cases, while others may be effective in reducing transmission [7]. Besides, large-scale vaccination aimed at achieving herd immunity poses many logistic and social difficulties [20], with diverse rational designs [21, 22], and prioritization plans of vaccination determining the evolution of the COVID-19 pandemic. The willingness of the population to get vaccinated and the economic costs, considering that free vaccines are not always available in many countries, are also variables to acknowledge when designing and analyzing the impacts of vaccination campaigns [23, 24]. A successful and equitable vaccination strategy will obviously have to carefully consider each one of these points, and some fundamental ethical choices already agreed upon.

The aim of the present work is to use an epidemiological model, fitted from real past data, to discuss the effectiveness of different vaccination strategies, and particularly the role of individual with a significantly larger number of contacts in the spread of the virus, called here "superspreaders" (not to be confused with individuals with a higher virus shedding), as depicted in Fig 1. Possible superspreaders comprise teachers at all levels, public transport and supermarket workers dealing directly with the public, among other social network hubs. This

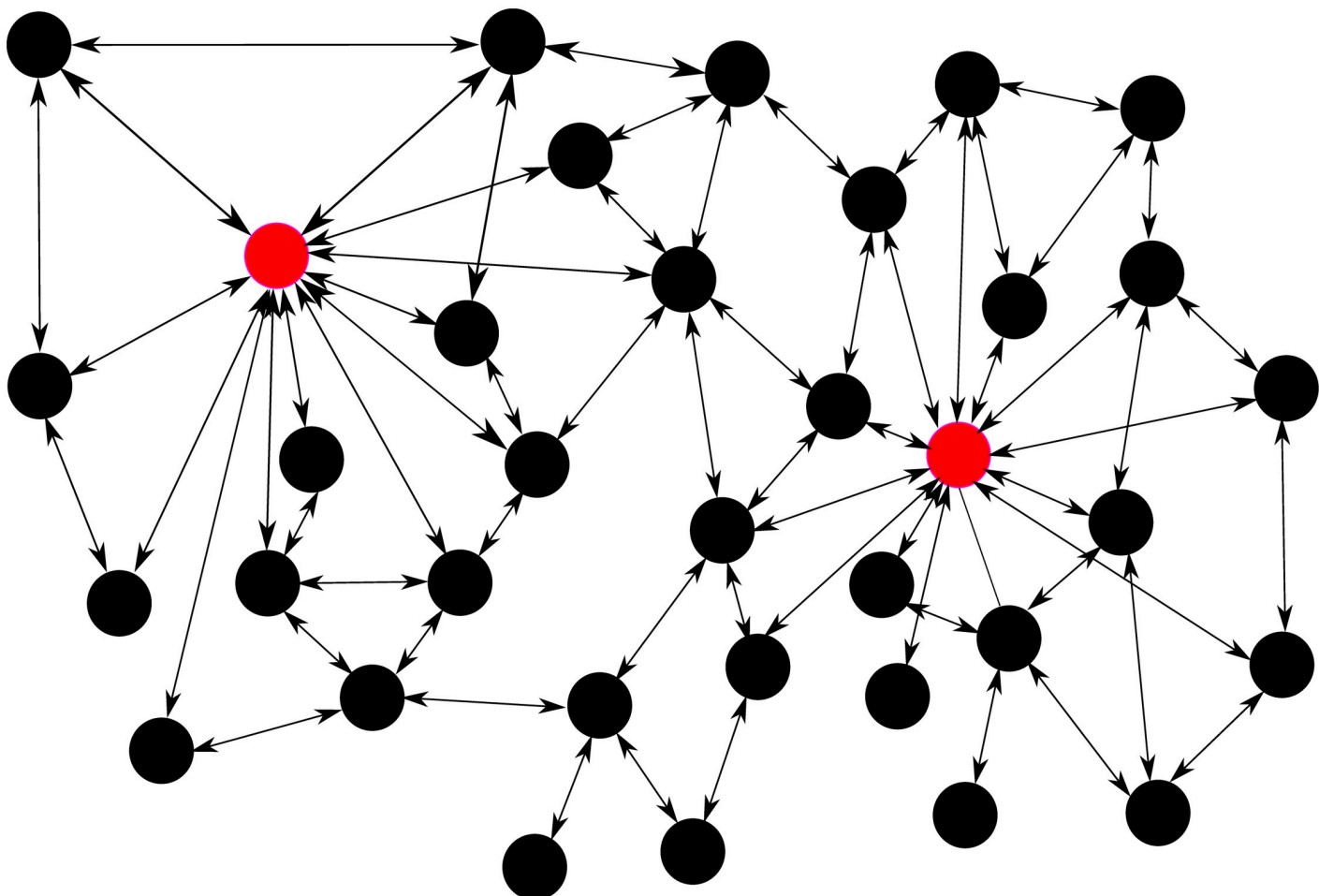

**Fig 1. Pictorial representation of contact hubs: Superspreaders (in red) have a much higher number of contacts than the average individuals (in black).** Arrows represent social contacts able to transmit the virus.

type of population heterogeneity can have relevant effects on the spread of the pandemic and should be considered carefully when designing a vaccination strategy. Indeed, previous works on fictional communities arranged in free-scale networks show that the choice of who should be vaccinated first can greatly impact the evolution of an epidemic [25–30].

We do not intend to reproduce current vaccination strategies or predict future outcomes given the current situation, but to discuss different hypothetical scenarios and discuss how future vaccination campaigns can be optimized in light of the results presented here. We chose to analyze the cases of Brazil and Portugal. Besides being the countries of the authors and historically very connected, they were both heavily inflicted by the still on-going COVID-19 pandemic, have very different sizes and populations, and with quite different vaccination stages, with 20.3% and 64.75% of the population in Brazil and Portugal fully vaccinated, respectively [18, 31]. This allows for a more thorough comparison of the proposed scenarios with the known current situation in both countries. On the other hand, many other countries are still in a very early stage of vaccinating their populations, mainly in Africa, where many countries have fully vaccinated less than 2% of its whole population [32]. We hope that the results presented here will contribute to the future design of vaccination strategies.

## Methods

### Epidemiological model

We consider an age-stratified SEIAHRV model with homogeneous mixing and compartments for susceptible (S), exposed (E), symptomatic infected (I), asymptomatic infected (A), hospitalized (H), recovered (R), vaccinated without primary vaccination failure (V) and vaccinated with primary vaccination failure (U) individuals, as described in Table 1. The age groups considered are 0 to 9, 10 to 19, 20 to 29, 30 to 39, 40 to 49, 50 to 59, 60 to 69, 70 to 79 and 80 or more years of age. All variables in the model are proportions with respect to the initial population $N_0$ (the current population changes due to mortality and birth). The flow chart for the model is given in Fig 2 and the corresponding differential equations in Eq. (S1) of the S1 File. All required parameters are given in [8, 33–36] and are shown in S1-S3 Tables in S1 File.

The force of infection $\lambda_i$ in Fig 2 for the $i$-the age group is given by

$$\lambda_i = \sum_{j=1}^{M} \beta_{i,j}(I_j + \xi A_j)/n_i, \tag{1}$$

where $\beta_{i,j}$ are the components of the transmission matrix giving the probability per unit of time that a symptomatic infected individual ($I_j$) of age group $j$ to infect a susceptible individual ($S_i$) of age group $i$. For an asymptomatic individual ($A_j$) this probability is $\chi\beta_{i,j}$. The transmission matrix is related to the contact matrix by the relation $\beta_{i,j} = p_c C_{i,j}$ where $p_c$ is the probability of contagion of a susceptible individual by an infectious symptomatic individual and supposed to be age-independent. The contact matrix varies with time due to social distancing and behavioral changes during the pandemic. We suppose here that such time variation can be represented by a single time-dependent factor $\omega(t)$ such that $C_{i,j}(t) = \omega(t)C_{i,j}(0)$, with $C_{i,j}(0)$ the components of the contact matrix prior to the pandemic, and thus $\beta_{i,j} = P(t)C_{i,j}(0)$ with $P(t) \equiv \omega(t)p_c$.

The contact matrix can be determined from ad-hoc suppositions and by some type of fitting of incidence data in a given population [37, 38]. A more realistic estimation can be obtained from field work recording the contacts of a sample of individuals of different age groups as implemented for a few countries: eight European countries (Belgium, Germany, Finland, Great Britain, Italy, Luxembourg, The Netherlands and Poland) [47], China [39], France [40], Japan [41], Kenya [42], Russia [43], Uganda [44], Zimbabwe [45] and Hong Kong [46]. We estimate the contact matrix using the average value of contacts from Mossong et al. [47] (see additional discussion in the S1 File).

**Table 1. Variables in the epidemiological model.** All proportions are with respect to the initial population $N_0$. The index $i = 1, \ldots, M$ denotes the age group ($M = 9$ in the present case).

| Variable | Description |
|---|---|
| $S_i$ | Proportion of susceptible individuals |
| $E_i$ | Proportion of exposed individual in the incubation period and not contagious. |
| $I_i$ | Proportion of infected symptomatic individuals (contagious). |
| $A_i$ | Proportion of infected asymptomatic individuals (contagious). |
| $H_i$ | Proportion of hospitalized individuals. |
| $R_i$ | Proportion of recovered individuals. |
| $V_i$ | Proportion of vaccinated individuals without primary vaccination failure. |
| $U_i$ | Proportion of vaccinated individuals with primary vaccination failure. |
| $n_i$ | Proportion of the population in the $i$-th age group. |

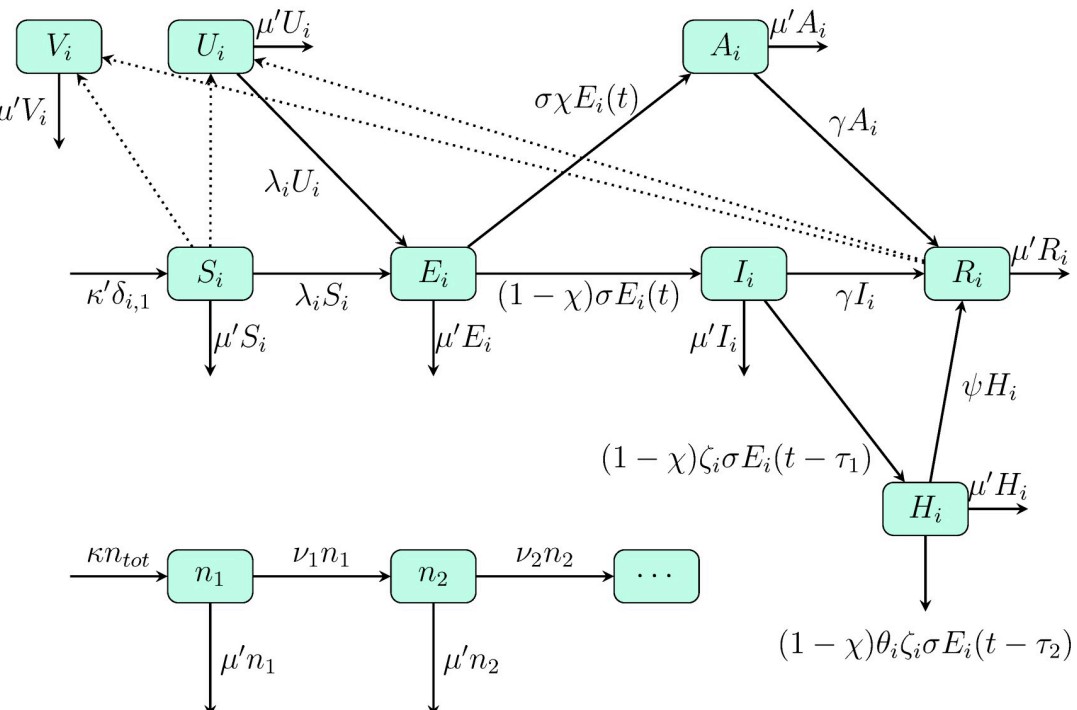

**Fig 2. Epidemiological SEIAHRV model flow chart.** The continuous arrows represent rates between variables. The dotted lines indicate the proportion with respect to $N_0$ of vaccine shots (see below). In the diagram $\mu' \equiv \mu N/N_0$ and $\kappa' \equiv \kappa N/N_0$, with $\mu$ and $\kappa$ the natural death and birth rates in the population, respectively, and $N$ the current total population. The proportion in the age group $i$ with respect to $N_0$ is denoted by $n_i$ and the aging rate $\nu_i$ from group $i$ is given by the inverse of the time span of the age group in the time unit used. All parameters are given in S1-S3 Tables in S1 File.

## Implementing the superspreaders group

Alongside the nine age groups defined above, we introduce a tenth group, which we call here superspreaders, composed by individuals with a much higher number of social contacts than the average in the population. Those are usually economically active individuals such as personnel in retail shops, public transport workers, health care personnel and teachers, among others. As a first simpler approach, and noting that the average age of such individuals are in the thirties, we consider that 20% of the population in the age group of 30 to 39 years old (the fourth age group) as superspreaders, with the same epidemiological parameters but different number of contacts per unit of time. This percentage is of course arbitrary and certainly varies from place to place, but it is a rough approximation of the active population with jobs requiring a large number of social contacts. The new contact matrix $\overline{C}$ can then be written as

$$\overline{C}_{i,j} = C_{i,j}, \quad i = 1, \ldots, M, \ i \neq 4, \ j = 1, \ldots, M,$$
$$\overline{C}_{4,j} = 0.8 \, C_{4,j}, \quad j = 1, \ldots, M, \tag{2}$$
$$\overline{C}_{i,M+1} = \alpha C_{i,4}, \quad \overline{C}_{i,M+1} = \alpha C_{i,4}, \quad i = 1, \ldots, M, \quad \overline{C}_{M+1,M+1} = \alpha C_{4,4},$$

where $\alpha$ is a contact factor denoting the excess in contacts with respect to the average of the 30–39 years age group. Supposing that with implemented measures for social distancing $\alpha$ is kept at the value $\alpha = 1$ and at a later stage, mimicking a return to normal activities, it is set to a value $3 \leq \alpha \leq 10$.

Population per age group in Brazil is obtained from the 2010 census data [48], corrected from official estimates for the population in each Brazilian state and the Federal District and available at [48]. Current data for Portugal is available at [49]. In the present work Portugal is considered as a whole, and model parameters for Brazil are separately fitted for each of the 26 states plus the Federal District, and final results are then added to obtain a gross total for the country. COVID-19 data for Portugal was obtained from the World Health Organization Coronavirus Disease (COVID-19) Dashboard [50], and data for each Brazilian state from the Brazilian Health Ministry [51].

It is a well-known fact that the total number of cases is highly underestimated, mainly due to a limited number of tests, and that deaths by COVID-19 are more reliable, although also subject to some under-reporting [52, 53]. As a consequence, fitting the model using the data series for the number of deaths yields results closer to the real situation. The transmission matrix is then obtained by fitting the function $P(t)$ in order to reproduce the reported time series of deaths (see S1 File).

## Results

We apply the epidemiological model described in the Methods section for Brazil and Portugal. In the present work, Portugal is considered as a whole, while model free parameters for Brazil are separately fitted for each of the 26 states plus the Federal District, and final results are then added to obtain a gross total for the country. The model is calibrated by fitting the model output for the cumulative total number of new deaths. Population data for both countries is available at [48] for Brazil and [49] for Portugal. The 2010 census data for Brazil is linearly update using the current population official estimates. Data for the total number of cases and deaths by COVID-19 is available for Portugal at [20] and for Brazil at [21]. The time series used here span the period from the first COVID-19 case detected (3/1/2020 for Portugal and 2/26/2020 for Brazil) up to August, 15 for both countries and we keep the stage of transmission fitted in the model corresponding for the dates of January, 15 and January, 1[st] for Brazil and Portugal, respectively. These are also the dates for the beginning of the vaccination scenarios we discuss here.

As a simpler approach to model the effects of a superspreader group we consider that 20% of the age group from 30 to 39 years of age are superspreaders, which amounts to 3.2% of the total population of Brazil and 2.5% of Portugal. We also suppose that due to social distancing, superspreaders follow the same contact pattern as other individuals of the same age group, and that at a further time (March, 16 2021 for Portugal and March 6, 2021 for Brazil), they resume full contact. We consider superspreaders as having $\alpha = 3$ to $\alpha = 10$ times more contacts in average that the number of contacts for the 30 to 39 years group, with $\alpha$ called the contact factor.

The population is split in the following age groups: 0 to 9, 10 to 19, 20 to 29, 30 to 39, 40 to 49, 50 to 59, 60 to 69, 70 to 79 and 80 years or more. We consider two different vaccine efficiencies: $e_v = 0.7$ and $e_v = 0.95$, and the following scenarios:

- Evolution without any vaccination;

- Vaccination plan 1: First vaccinate individuals with 60 years of age and older, and then those from other age groups in descending order of age;

- Vaccination plan 2: Start vaccination by superspreaders, and then follow the order as in the previous case.

We also assume that vaccines protect against the disease and avoid transmission, that full immunization is attained after 30 days of the first dose and that two doses are required. The

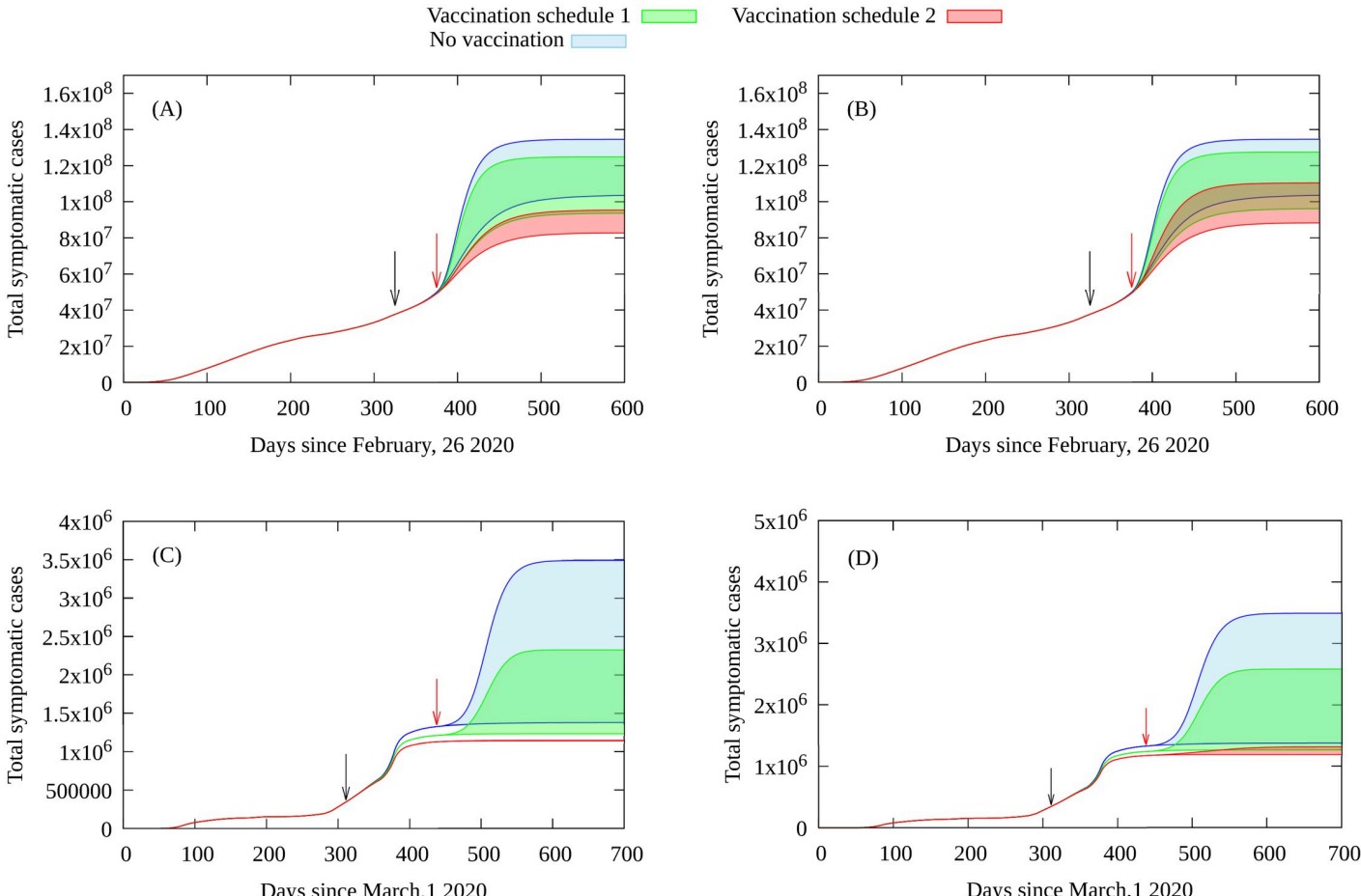

**Fig 3. Total number of deaths obtained from the fitted epidemiological model.** The results are for two different vaccine efficacy scenarios for Brazil A) $e_v$ = 95%, B) $e_v$ = 70%, and for Portugal with C) $e_v$ = 95%, D) $e_v$ = 70%. Black arrows indicates the beginning of each vaccination plan while red arrows the moment when superspreaders return to full activity, with $\alpha$ = 3 to $\alpha$ = 10 times more contacts than typically for its age group (30 to 39 years old). The shaded blue area gives the prognostics in the absence of any vaccination. The green area gives the prognostics with vaccination starting with individuals aged 60 years and older and then vaccinating the remaining population in descending order of age group (vaccination plan 1). The red region corresponds to vaccination plan 2 starting with the superspreaders and then proceeding in the same order as in vaccination plan 1. Each shaded region is delimited by the evolution with contact factors $\alpha$ = 3 and $\alpha$ = 10.

total supposed number of doses is 20 million for Portugal and 250 million for Brazil, available in a time span of one year. The predicted number of deaths for each scenario from our epidemiological model is given in Fig 3, and the total number of cases in Fig 4. For details on the model and it is calibrated from empirical data see the Methods section and S1 File.

Healthcare demand is also estimated from hospitalized variable $H_i$ in the model. The proportions of mild, severe and critical COVID-19 cases among symptomatic individuals are 80.9%, 13.8% and 4.7%, respectively [11], and we assume that severe and critical cases require hospitalization and that all severe cases demand ICU attention, i.e. 25.4% of the hospitalized individuals [11]. The estimated ICU demand in beds for 2021 in each hypothetical scenario, are shown in Table 2.

The total number of symptomatic cases for each vaccination (and no vaccination) scenario are shown in Fig 4. By including the superspreaders in the first group to be vaccinated results in a significant reduction in the number of COVID-19 cases, which at its turn results in reduction in hospitalizations. We note that this reduction in the total number of symptomatic cases

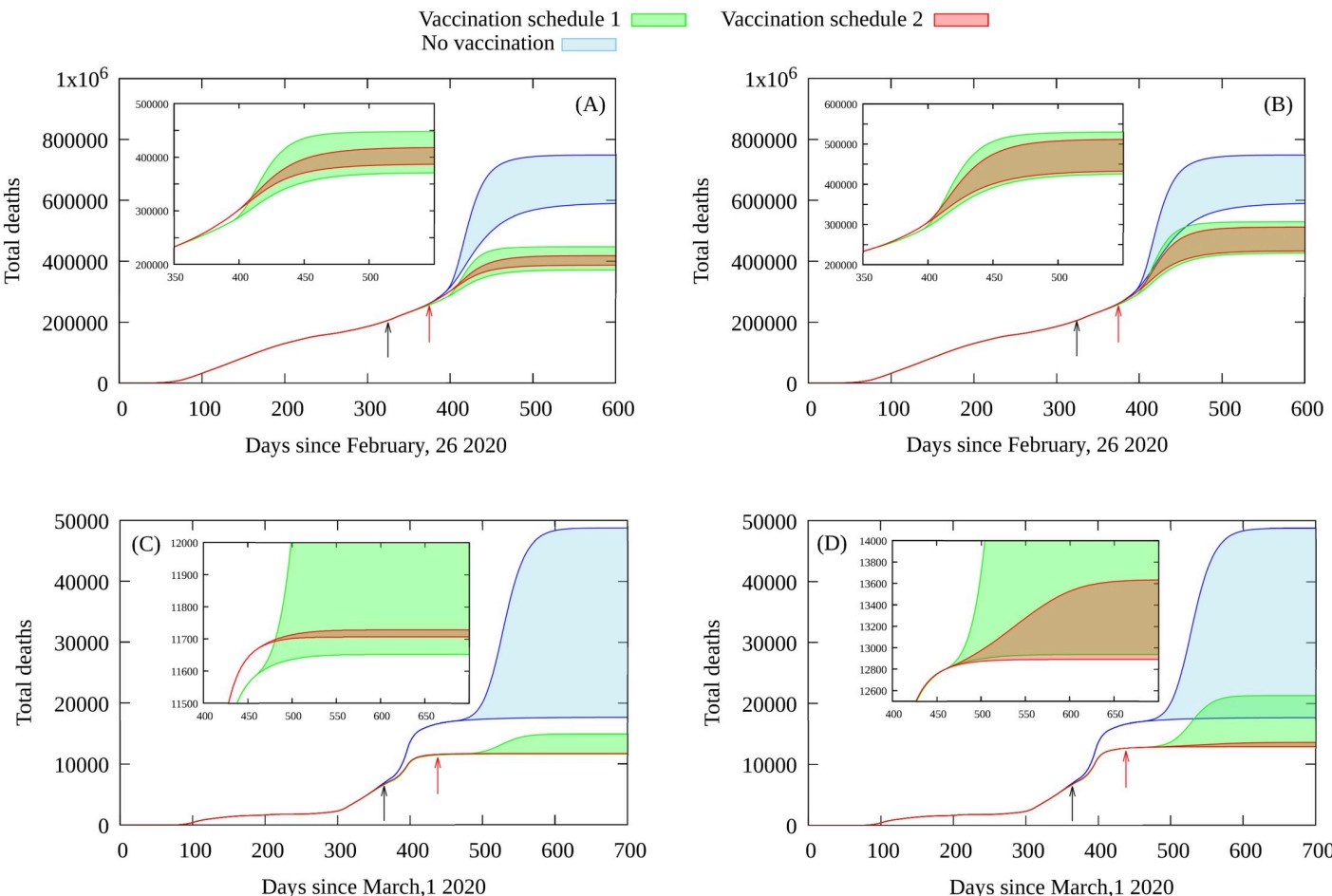

**Fig 4. Total number of symptomatic cases: Brazil (A)** $e_v = 0.95$, **(B)** $e_v = 0.7$, **and Portugal (C)** $e_v = 0.95$ **and (D)** $e_v = 0.7$, **for the vaccinations campaigns with superspreaders as described in the main text.** The black arrows indicate starting of vaccination and red arrows the moment superspreaders return to full social contact ($3 \leq \alpha \leq 10$). The lower and upper curves defining the green and red shaded areas correspond to $\alpha = 3$ and $\alpha = 10$, respectively.

**Table 2. Estimated demand of ICU beds for each vaccination plan and limit contact factors $\alpha = 3$ and $\alpha = 10$.**

| Plan 1 | | | | | plan2 | | | |
|---|---|---|---|---|---|---|---|---|
| Country | $\alpha$ | $e_v$ | ICU beds | | Country | $\alpha$ | $e_v$ | ICU beds |
| Brazil | 3 | 0.7 | 74490 | | Brazil | 3 | 0.7 | 63339 |
| | 3 | 0.95 | 66720 | | | 3 | 0.95 | 52966 |
| | 10 | 0.7 | 179754 | | | 10 | 0.7 | 117786 |
| | 10 | 0.95 | 162049 | | | 10 | 0.95 | 73659 |
| Portugal | 3 | 0.7 | 1990 | | Portugal | 3 | 0.7 | 1835 |
| | 3 | 0.95 | 1811 | | | 3 | 0.95 | 1619 |
| | 10 | 0.7 | 2179 | | | 10 | 0.7 | 1835 |
| | 10 | 0.95 | 1811 | | | 10 | 0.95 | 1619 |

also reduces the number of individuals with long-term health effects due to COVID-19, also resulting in a reduction in health spending in each country.

## Discussion

We note that starting vaccination by the superspreader group, as defined here, is only effective if their average contact number is a few times that of the average in the population. Using the estimated contact matrix, we obtained the average number of contacts of a single individual with individuals of any age group in Brazil as 15.6 per day and 13.8 per day in Portugal. This implies that the superspreader group has approximately 4 to 12 times the average number of contact in the population as obtained from the entries of the contact matrix (see S1 File), for the contact factor range of $\alpha = 3$ to $\alpha = 10$. From our results we observe a threshold value for the contact factor such that starting vaccination by the superspreaders followed by the eldest members of the population is more beneficial, in the sense that the death toll and the total number of cases decrease significantly if compared to vaccination starting only by the eldest, as seen in Fig 3. This also depends on the current situation of the pandemic in each location, and is more pronounced in an expanding phase. The reduction in the peak demand for ICU beds in Table 2 is also significant, an important point in order to avoid the overwhelming of health facilities.

From the contact matrix one could argue that first vaccinating those in the age group of 10 to 19 years of age, the one with the naturally highest number of contacts due to school attendance, could reduce the overall virus transmission. While this is true, the time span required would leave the eldest age group exposed to the virus, resulting in a higher overall mortality. Finally, the simpler case considered here of superspreaders being limited to the 30–39 years age group can be extended to encompass other age groups, by taking into account demographics and data on occupation distribution for each population, to obtain a more realistic estimate of the superspreader group and the most beneficial vaccination strategy.

We note that the boundaries of each simulated scenario for plan 2 (given by $\alpha = 3$ and $\alpha = 10$), represented by the shaded regions in red in Fig 3, have a smaller width than the one for plan 1 (in green). This implies a weaker dependency on the number of contacts of the superspreaders, which is another favorable point for first vaccinating the superspreaders group. We also considered the important issue of the expected number of ICU beds required. For the scenarios considered here it is significantly smaller if superspreaders are vaccinated first, with an the overall reduction of the number of persons requiring hospitalization for any value of contact factor considered here. These results suggest that starting vaccination by those with a a much greater number of contacts allows a greater flexibility in economic activities, due to the smaller dependent on the size of superspreaders group and their contact structure.

A more detailed study for each location using data on the economic activities, if available, is necessary to design a more efficient vaccination plan in the line discussed here, and increase the overall number of lives saved, and given the current conditions and considering the lack of basic resources for vaccination in several countries. Our results can be extended straightforwardly to other countries and show the benefits of a carefully designed vaccination strategy maximizing the results from a limited number of vaccine doses and limited infrastructure and logistics. We also note that the predicted number of hospitalized individuals correspond to an ideal setting, such that every severe or critical case will be hospitalized. This is not always the case due to limitations of available medical facilities. Therefore, a reduction of the hospitalization demand is also critically important for a reduction in the total number of deaths caused by the disease.

More than 50% of the population of both countries considered in the present study are already fully vaccinated, but the considerations here are still valid for other groups in the population such as children and adolescents who have an high number of contacts due to school attendance and are not yet fully vaccinated, mainly in Brazil. The main objective of the present study is to present a proof of concept for the importance of different vaccination strategies than those currently considered in most of the world, and draw attention to those individuals with a very high number of contacts, which are superspreaders in the sense that, beside being more prone to infection, they also function as hubs for the propagation of the virus. Using a modeling approach allows us to explore possible different scenarios and bring forward the need for a more thorough investigation into the optimal use of vaccines, particularly in those countries which are still lagging behind in the proportion of the population already fully vaccinated.

## Limitations

As any model based approach, the results can only reflect what is fed into the model itself. Our approach relies on epidemiological parameters reported in the literature, which can vary for different localities and over time, e. g. the infection fatality ratio due to the overwhelming health infrastructure, partially considered for Brazil, or new variants. We also supposed that the reinfection proportion is negligible and that mixing is homogeneous, discarding heterogeneous effects in the population. The identification of superspreaders is also a difficult element of our approach, although some professional activities can be readily recognized as such, as public transport workers, health care personnel and teachers.

## Supporting information

**S1 File.**
(ZIP)

## Author Contributions

**Conceptualization:** Tarcísio M. Rocha Filho.

**Formal analysis:** Tarcísio M. Rocha Filho, José F. F. Mendes, Thiago B. Murari, Aloísio S. Nascimento Filho, Antônio J. A. Cordeiro, Walter M. Ramalho, Fúlvio A. Scorza, Antônio-Carlos G. Almeida, Marcelo A. Moret.

**Investigation:** José F. F. Mendes.

**Methodology:** Tarcísio M. Rocha Filho, José F. F. Mendes, Thiago B. Murari, Fúlvio A. Scorza, Antônio-Carlos G. Almeida, Marcelo A. Moret.

**Software:** Tarcísio M. Rocha Filho.

**Writing – original draft:** Tarcísio M. Rocha Filho, José F. F. Mendes, Aloísio S. Nascimento Filho, Antônio J. A. Cordeiro, Walter M. Ramalho, Fúlvio A. Scorza, Antônio-Carlos G. Almeida, Marcelo A. Moret.

**Writing – review & editing:** Tarcísio M. Rocha Filho, José F. F. Mendes, Thiago B. Murari, Aloísio S. Nascimento Filho, Antônio J. A. Cordeiro, Walter M. Ramalho, Antônio-Carlos G. Almeida, Marcelo A. Moret.

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
