## [Decision Letter · Decision Letter 0]

6 Oct 2021

PONE-D-21-27433Optimization of COVID-19 vaccination and the role of individuals with a high number of contacts: a model based approachPLOS ONE

Dear Dr. 33457670153,

Thank you for submitting your manuscript to PLOS ONE. After careful consideration, we feel that it has merit but does not fully meet PLOS ONE’s publication criteria as it currently stands. Therefore, we invite you to submit a revised version of the manuscript that addresses the points raised during the review process.

ACADEMIC EDITOR: The following major issues must be addressed: Could you justify why the particular age group was picked for the model development?The authors only chose Brazil and Portugal. They are in different phases of vaccination. In sensing the full data, my first notice is that Brazil fitting into the unpredictable model is just 20.3 percent compared to Portugal. It would be amazing if the authors recognized that the vaccination stage is over 50% or more other than Brazil and Portugal.I have noticed some typos and grammatical issues, thus read the article carefully.Please submit your revised manuscript by Nov 20 2021 11:59PM. If you will need more time than this to complete your revisions, please reply to this message or contact the journal office at plosone@plos.org. Please include the following items when submitting your revised manuscript:A rebuttal letter that responds to each point raised by the academic editor and reviewer(s). You should upload this letter as a separate file labeled 'Response to Reviewers'.A marked-up copy of your manuscript that highlights changes made to the original version. You should upload this as a separate file labeled 'Revised Manuscript with Track Changes'.An unmarked version of your revised paper without tracked changes. You should upload this as a separate file labeled 'Manuscript'.

We look forward to receiving your revised manuscript.

Kind regards,

M. Shamim Kaiser, PhD

Academic Editor

PLOS ONE

Journal Requirements:

"17474"

"This work received financial support from the National Council of Technological and Scientific 222

Development - CNPq (grant numbers 302449/2019-1 FAS, 309617/2020-0 ACGA, 223

305291/2018-1 MAM), Bahia State Research Support Foundation (BOL0723/2017 AJAC) 224

(Brazil) and i3N (grant numbers UIDB/50025/2020 & UIDP/50025/2020) - Fundação para a 225

Ciência e Tecnologia/MEC (Portugal)"

"17474"

5. Please upload a copy of Figure S3, to which you refer in your text on pages 5 and 6. If the figure is no longer to be included as part of the submission please remove all reference to it within the text.

Reviewers' comments:

Reviewer's Responses to Questions

**Comments to the Author**

1. Is the manuscript technically sound, and do the data support the conclusions?

Reviewer #1: Yes

Reviewer #2: Yes

2. Has the statistical analysis been performed appropriately and rigorously? 

Reviewer #1: Yes

Reviewer #2: Yes

3. Have the authors made all data underlying the findings in their manuscript fully available?

Reviewer #1: Yes

Reviewer #2: Yes

4. Is the manuscript presented in an intelligible fashion and written in standard English?

Reviewer #1: Yes

Reviewer #2: No

5. Review Comments to the Author

Reviewer #1: This manuscript is well written but needs correction of some important points before publication. Like in introduction need to add information from the below recent papers;

https://doi.org/10.3390/vaccines9080864

https://doi.org/10.3390/vaccines9050416

There is not enough methodological information in the abstract.

Present the methods section before the results and discussion.

Could you please justify the reasons why the specific age group was selected for the model development and what are the reasons for vaccination failure among that group.

Add more clarity around super spreaders group.

The strengths of the research are not clearly established.

Reviewer #2: The research ‘Optimization of COVID-19 vaccination and the role of individuals with a high number of contacts: a model based approach’ is interesting. It proposed a methodology to carefully analyze the current situation of Covid-19, considering the social contacts structure in a given population and its age distribution. Alongside, it deeply analyzed the vaccine efficacy, prioritizations of vaccination strategies, successful and equitable vaccination strategy and vaccine input methodology by fitting different hypothetical scenarios into a epidemiological model. Its not a big surprise that super-spreaders play the key role for transmission, infection, fatality as well as prevention of any contagious diseases like covid-19. But the way the authors use the data of super-spreaders in the model is a good one.

The authors have chosen only Brazil and Portugal as they are originated from these countries. These they are in different vaccination stages, with 20.3% and 64.75% of the population in Brazil and Portugal fully vaccinated, respectively. By sensing the whole study, my first observation is that the case of Brazil fitting in the model given unseemly results in compare with Portugal as Brazil has only 20.3%. Its troublesome to discuss future optimal vaccination campaigns for any community by model fitting only a small population like Brazil. It would be great if the authors considered another besides Brazil and Portugal where the vaccination stage is above 50% or more.

How does the authors deal with the uncertainties of the model. There are many considerations the authors considered to run the model. For example, reinfection proportion is negligible and that mixing is homogeneous, discarding heterogeneous effects in the population. Why and How?

How does the authors consider that 20% of the age group from 30 to 39 years of age are superspreaders, which amounts to 3.2% of the total population of Brazil and 2.5% of Portugal?

Different age groups (9 groups here) should have different behavioral pattern and it varies with locality and time. Why didn’t the authors use any coefficient or else to remove the uncertainties raised the by the said issues?

In a study I found that prioritizing adults aged 60+ years remained the best way to reduce mortality and YLL for R0 ≥ 1.3, but prioritizing adults aged 20 to 49 years was superior for R0 ≤ 1.2. Prioritizing adults aged 20 to 49 years minimized infections for all values of R0 investigated. How does this results connect with the results in this manuscript. If differ, why?

It would be good if the authors considered two factors more e.g. vaccines with imperfect transmission-blocking effects, incorporation of population seroprevalence and individual serological testing.

I found many works in the literature on approximately same topic. In such context, this can be considered a prototype work. But, the model extracted results offered herein this study provoked me to the favor of publishing this manuscript in Plos One. In my opinion, it needs a further review after a major revision where the authors address the above flaws.

6. PLOS authors have the option to publish the peer review history of their article (what does this mean?). If published, this will include your full peer review and any attached files.

Reviewer #1: No

Reviewer #2: No

---

## [Author Response · Author response to Decision Letter 0]

19 Nov 2021

ACADEMIC EDITOR: 

The following major issues must be addressed:

COMMENT: "Could you justify why the particular age group was picked for the model development?"

ANSWER: The different age groups were chose based on the available information by age group on hospitalization and death probabilities by infection. The age group for superspreaders correspond to the the "average" age group pf the economically active population, and is a simplification introduced in the model. In this way, we believe it depicts well what occurs with those individuals with a much greater number of contacts in the whole population.

COMMENT: "The authors only chose Brazil and Portugal. They are in different phases of vaccination. In sensing the full data, my first notice is that Brazil fitting into the unpredictable model is just 20.3 percent compared to Portugal. It would be amazing if the authors recognized that the vaccination stage is over 50% or more other than Brazil and Portugal."

ANSWER: We added a comment on this in the last paragraph of the Discussion section.

COMMENT: "I have noticed some typos and grammatical issues, thus read the article carefully."

ANSWER: we did it.

Journal Requirements:

We correctec the funding information in the Acknowledgments section. We added to it the phrase "The funders had no role in study design, data collection and analysis, decision to publish, or preparation of the manuscript."

COMMENT: "5. Please upload a copy of Figure S3, to which you refer in your text on pages 5 and 6. If the figure is no longer to be included as part of the submission please remove all reference to it within the text."

ANSWER: There is no citation to Figure S3, but to Table S3, which is included in the supplemental material.

COMMENT: "PLOS authors have the option to publish the peer review history of their article (what does this mean?). If published, this will include your full peer review and any attached files."

ANSWER: Yes

Reviewers' comments:

Reviewer #1

COMMENT: "This manuscript is well written but needs correction of some important points before publication. Like in introduction need to add information from the below recent papers;

https://doi.org/10.3390/vaccines9080864

https://doi.org/10.3390/vaccines9050416"

ANSWER: We added a comment on the need to consider willingness to vaccinate and the economic cost for individuals, and cited the suggested references.

COMMENT: "There is not enough methodological information in the abstract."

ANSWER: We added some information of the methodology in the abstract

COMMENT: "Present the methods section before the results and discussion."

ANSWER: Done.

COMMENT: "Could you please justify the reasons why the specific age group was selected for the model development and what are the reasons for vaccination failure among that group.

Add more clarity around super spreaders group."

ANSWER: We explained more explicitly in the first paragraph of the subsection "Implementing the superspreaders group".

COMMENT: "The strengths of the research are not clearly established."

ANSWER: We added a paragraph in the end of the Discussions section. We believe that this will put more clearly the strength and goals of

the present work.

Reviewer #2

COMMENT: "The authors have chosen only Brazil and Portugal as they are originated from these countries. These they are in different vaccination stages, with 20.3% and 64.75% of the population in Brazil and Portugal fully vaccinated, respectively. By sensing the whole study, my first observation is that the case of Brazil fitting in the model given unseemly results in compare with Portugal as Brazil has only 20.3%. Its troublesome to discuss future optimal vaccination campaigns for any community by model fitting only a small population like Brazil. It would be great if the authors considered another besides Brazil and Portugal where the vaccination stage is above 50% or more."

ANSWER: The present work is a first study in this direction, and is a proof of concept for the need of a more detailed consideration of different possibilities for vaccination strategies. A more detailed study, with a version of the model considering different kinds of vaccine is under way, mainly for countries with a very low proportion of the population fully vaccinated and located mainly in sub-Saharan Africa. We hope that the current approach may be useful in countries where the available number of doses is still very small.

COMMENT: "How does the authors deal with the uncertainties of the model. There are many considerations the authors considered to run the model. For example, reinfection proportion is negligible and that mixing is homogeneous, discarding heterogeneous effects in the population. Why and How?"

ANSWER: As for any modeling approach, some simplifications are considered, while more important aspects are considered with more care. Heterogeneous effects may be considered in more complex models, but are also very hard to model as more information is needed. The use of homogeneous mixing, although being a simplification often used, allows to calibrate the model with available data and yet obtain important insights on the pandemic dynamics. Reinfection is expected to be not significant for the amount of time considered in the present work.

COMMENT: "How does the authors consider that 20% of the age group from 30 to 39 years of age are superspreaders, which amounts to 3.2% of the total population of Brazil and 2.5% of Portugal?"

ANSWER: We introduced some explanation on this on the first paragraph of the subsection on "Implementing the superspreaders group".

COMMENT: "Different age groups (9 groups here) should have different behavioral pattern and it varies with locality and time. Why didn’t the authors use any coefficient or else to remove the uncertainties raised the by the said issues?"

ANSWER: This is indeed true. The dependence on locality is obtained mainly by field studies such as the one in Mossong et al. Unfortunately no such study was performed for Brazil or Portugal, so we have to assume that the average number of contacts between two age-groups is the same as the average for the European countries considered by Mossong et al. This a a quite reasonable assumption as both Brazil and Portugal have mainly an European culture and most of its population live in cities, as in most Europe. Besides the contact matrix used in the model is obtained properly considering the population by age-group in each country, which results in different contact matrices. The time evolution of the contact structure is approximately taken into account by a constant factor multiplying the contact matrix that considers at the same time the increase or decrease of average contacts and probability of transmission, and fitted from real data, as explained in the text.

COMMENT: "In a study I found that prioritizing adults aged 60+ years remained the best way to reduce mortality and YLL for R0 ≥ 1.3, but prioritizing adults aged 20 to 49 years was superior for R0 ≤ 1.2. Prioritizing adults aged 20 to 49 years minimized infections for all values of R0 investigated. How does this results connect with the results in this manuscript. If differ, why?"

ANSWER: The point here is that if you have to chose only by age-group, then it starting by the group with higher age results in the most significant reduction of hospitalizations and deaths, and this is currently the most frequent strategy adopted by far. What we are drawing some attention on that there are other possibilities that may result in a more significant reduction of cases and deaths by COVID-19, i.e. by first vaccinating the superspreaders group and afterwards by following reverse chronological order of age.

COMMENT: "It would be good if the authors considered two factors more e.g. vaccines with imperfect transmission-blocking effects, incorporation of population seroprevalence and individual serological testing."

ANSWER: This was partially taken into consideration when considering vaccines with an efficiency varying from 0.7 to 0.95. A more detailed study in this direction is underway in out group.

COMMENT: "I found many works in the literature on approximately same topic. In such context, this can be considered a prototype work. But, the model extracted results offered herein this study provoked me to the favor of publishing this manuscript in Plos One. In my opinion, it needs a further review after a major revision where the authors address the above flaws."

ANSWER: We believe that the above mentioned corrections and answers satisfy this referee's comment.

---

## [Decision Letter · Decision Letter 1]

26 Dec 2021

Optimization of COVID-19 vaccination and the role of individuals with a high number of contacts: a model based approach

PONE-D-21-27433R1

Dear Dr. 33457670153,

We’re pleased to inform you that your manuscript has been judged scientifically suitable for publication and will be formally accepted for publication once it meets all outstanding technical requirements.

Kind regards,

M. Shamim Kaiser, PhD

Academic Editor

PLOS ONE

Additional Editor Comments (optional):

Thank you for responding to all of the reviewers' concerns.

However, the paper contains a few mistakes. Please take your time reading this paper carefully.

Reviewers' comments:

Reviewer's Responses to Questions

**Comments to the Author**

1. If the authors have adequately addressed your comments raised in a previous round of review and you feel that this manuscript is now acceptable for publication, you may indicate that here to bypass the “Comments to the Author” section, enter your conflict of interest statement in the “Confidential to Editor” section, and submit your "Accept" recommendation.

Reviewer #1: All comments have been addressed

Reviewer #2: All comments have been addressed

2. Is the manuscript technically sound, and do the data support the conclusions?

Reviewer #1: Yes

Reviewer #2: Yes

3. Has the statistical analysis been performed appropriately and rigorously? 

Reviewer #1: Yes

Reviewer #2: Yes

4. Have the authors made all data underlying the findings in their manuscript fully available?

Reviewer #1: Yes

Reviewer #2: Yes

5. Is the manuscript presented in an intelligible fashion and written in standard English?

Reviewer #1: Yes

Reviewer #2: Yes

6. Review Comments to the Author

Reviewer #1: Thanks for doing all the changes as suggested. The manuscript sounds technically good and well revised.

Reviewer #2: (No Response)

7. PLOS authors have the option to publish the peer review history of their article (what does this mean?). If published, this will include your full peer review and any attached files.

Reviewer #1: No

Reviewer #2: No

---

## [Editor Report · Acceptance letter]

10 Jan 2022

PONE-D-21-27433R1 

Optimization of COVID-19 vaccination and the role of individuals with a high number of contacts: a model based approach 

Dear Dr. 33457670153:

I'm pleased to inform you that your manuscript has been deemed suitable for publication in PLOS ONE. Congratulations! Your manuscript is now with our production department. 

Kind regards, 

on behalf of

Dr. M. Shamim Kaiser 

Academic Editor

PLOS ONE